# Borax-based gel electrophoresis: A novel approach for RNA integrity analysis

Abdulhadi Albaser  *

Department of Microbiology, Faculty of Science, Sebha University, Sebha, Libya

* abd.albaser@sebhau.edu.ly

## Abstract

This study investigates the unexpected utility of borax-based agarose gel electrophoresis for RNA integrity assessment. Accurate RNA integrity analysis fundamentally requires denaturation to fully resolve RNA molecules from their complex secondary structures and prevent aggregation, a principle typically achieved with hazardous chemicals like formaldehyde. Through multiple independent experiments, this work demonstrates that borax, beyond its buffering properties, exhibits denaturing-like behavior, effectively separating RNA molecules with resolution comparable to formaldehyde-based methods. This finding is surprising and challenges conventional understanding. To validate the method's versatility, it was successfully applied to total RNA extracted from six diverse microbial species, including Gram-negative bacteria (*E. coli*, *Pseudomonas aeruginosa*), Gram-positive bacteria (*Staphylococcus aureus*, *Enterococcus faecalis*), and eukaryotic fungi (*Candida glabrata*, *Candida albicans*). This novel method offers a compelling and safer alternative to traditional formaldehyde-based approaches for RNA analysis in various research and clinical settings. It provides several advantages, including enhanced safety, simplified protocols, and reduced electrophoresis time. By eliminating the need for pre-treatment steps and utilizing borax as both a buffer and an apparent denaturant, this method significantly streamlines RNA analysis. While the precise mechanism underlying this denaturing-like effect requires further elucidation, this study highlights a promising, accessible, and universally applicable tool for robust RNA integrity assessment.

## Introduction

Maintaining RNA integrity is paramount for accurate gene expression analysis, reliable RT-PCR results, and successful RNA sequencing [1,2]. However, RNA is highly susceptible to degradation by RNases, ubiquitous enzymes found in diverse environments, including microorganisms, skin, and bodily fluids [3–11]. These enzymes, present in viruses [1,2], prokaryotes [3], and eukaryotes [4–6], pose a significant threat to RNA quality in both research and clinical settings. Proper RNA integrity assessment by gel electrophoresis fundamentally relies on denaturing conditions

**Data availability statement:** All relevant data are within the paper and its Supporting Information files.

**Funding:** The author(s) received no specific funding for this work.

**Competing interests:** The authors have declared that no competing interests exist.

to fully unfold RNA secondary structures. This ensures accurate migration based solely on size, leading to sharp resolution of ribosomal RNA bands (e.g., 23S and 16S in prokaryotes; 26S/28S and 18S in eukaryotes and clear visualization of degradation products. While simpler non-denaturing agarose gels can show RNA bands, they often fail to provide the necessary denaturation. Without it, RNA molecules can maintain complex secondary structures that cause aberrant migration, potentially leading to inaccurate sizing and obscuring subtle degradation that is crucial for comprehensive integrity assessment. Traditional methods for assessing RNA quality often employ hazardous chemicals such as formaldehyde and mercury hydroxide [12–15], posing significant safety risks and limiting accessibility, particularly in resource-constrained environments. This study presents a novel approach for RNA quality assessment using borax-based agarose gel electrophoresis. Borax, a readily available and cost-effective compound, offers distinct advantages over traditional methods [16–18]. By eliminating the need for hazardous chemicals like formaldehyde, it significantly enhances safety. The simplified protocol, which involves direct loading of RNA samples onto the gel without pre-treatment, reduces the time and effort required for RNA analysis. This method also offers the potential for reduced costs compared to some commercial reagents. This study demonstrates the feasibility and unexpected efficacy of using borax as both a buffer and an apparent denaturant for RNA electrophoresis, enabling rapid, efficient, and safe assessment of RNA integrity.

## Methods

### Gel preparation

A 0.8% agarose gel was prepared using a 5 mM sodium tetraborate decahydrate (borax) solution as the running buffer. For comparative analysis, a 0.8% agarose gel with standard TAE buffer was also prepared. Gel Red or ethidium bromide was added to both melted agarose solutions before casting.

### Sample preparation and RNA extraction

Total RNA samples were isolated from six distinct microbial species, including Gram-negative bacteria (*Escherichia coli ATCC 25922*, *Pseudomonas aeruginosa* [19]), Gram-positive bacteria (*Staphylococcus aureus*, *Enterococcus faecalis*), and eukaryotic fungi (*Candida glabrata*, *Candida albicans*). Bacterial and fungal strains were cultured to mid-log phase in appropriate liquid media. Total RNA was extracted using a modified, in-house developed acidic lysis protocol (pH 5.0), which eliminates the need for hazardous chemicals like phenol and costly enzymes. This specific protocol is detailed in Albaser (2026) [20]. For the purpose of this study, the high-quality RNA required for gel analysis was obtained by utilizing the following critical steps: The protocol was applied to all species (including *E. coli*) and involved: (1) Acidic Lysis: Cell pellets were mixed with a specialized acidic lysis buffer (pH 5.0) to keep RNA in solution; (2) Cell Disruption: A boiling step was introduced (2 minutes at 100°C) to effectively lyse the robust cell walls of Gram-positive bacteria and fungal species; (3) RNA Precipitation: Calcium chloride ($CaCl_2$) was

used as the precipitation agent, adapted to efficiently precipitate the target total RNA from the lysate while minimizing cellular contaminants. The extracted RNA was resuspended in RNase-free water and mixed with 1x colorless loading dye (or other specified variants) prior to loading. The purity was verified by spectrophotometry ($A_{260/280}$ ratio approx 1.95) before use.

## Electrophoresis

Approximately 20 µL (containing approximately 1.5–2.0 µg) of RNA sample mixed with 2 µL of 10X loading buffer (or other specified loading buffer variants) was loaded onto a 10 cm x 10 cm, 0.8% agarose gel. Electrophoresis was carried out at 120 V for 25–30 minutes for borax gels or 60 V for 60 minutes for TAE gels in a submerged gel electrophoresis system at room temperature.

## Loading buffer component experiment and preparation

To investigate the influence of various loading buffer components on RNA separation, the following 10X stock solution was prepared: 0.025 g bromophenol blue (BPB), 3.9 mL glycerol, 500 µL of 10% SDS, and 200 µL 0.5M EDTA, topped up to a final volume of 10 mL with $H_2O$. This resulted in a 10X stock concentration of 0.25% w/v BPB, 39% v/v glycerol, 0.5% w/v SDS, and 10 mM EDTA. Three distinct 10X loading buffer formulations were derived from this stock: (1) Standard Loading Buffer (full component stock); (2) SDS-Omitted Buffer (identical to standard but without SDS); and (3) Colorless Buffer (retaining Glycerol, SDS, and EDTA, but with the **omission of** Bromophenol Blue (BPB)). Total *E. coli* RNA samples were mixed 5:1 (RNA to buffer) and loaded directly onto the borax-based agarose gel.

## Concentration effect of Borax on RNA integrity

To investigate the concentration-dependent effects of borax on RNA integrity, RNA samples were incubated with a range of borax concentrations (50 µM - 10 mM) prior to loading. The treated samples were then loaded onto either borax or TAE gels and electrophoresed as described above.

# Results and discussion

### Traditional RNA electrophoresis methods

Traditional RNA electrophoresis methods often rely on formaldehyde as a denaturant, necessitating pre-treatment steps such as heating RNA samples with formaldehyde and subsequent cooling on ice. These steps are time-consuming and increase the risk of RNA degradation. Furthermore, formaldehyde is a hazardous chemical, posing potential health and environmental risks. This highlights the urgent need for safer and more efficient alternatives.

### Borax as an alternative buffer and apparent denaturant

Borax, a readily available and cost-effective compound, was investigated as an alternative buffer due to its known buffering capacity and ease of handling. This study explores the feasibility of using borax as an alternative buffer for RNA electrophoresis, aiming to develop a safer and more efficient method for RNA analysis. Remarkably, borax effectively separated RNA molecules, as evidenced by the clear and sharply resolved bands observed for 16S, 23S, and 5S rRNA in Fig 1. This observation strongly suggests that borax can effectively resolve RNA species, demonstrating its potential as an apparent denaturant for RNA electrophoresis without the need for traditional denaturants like formaldehyde. The sharp and distinct bands indicate high-quality RNA, comparable to those obtained with traditional denaturing gels (e.g., formaldehyde-based) and superior to standard non-denaturing gels for comprehensive integrity assessment. These results unequivocally demonstrate that borax serves as a viable and unexpected alternative to formaldehyde for RNA electrophoresis.

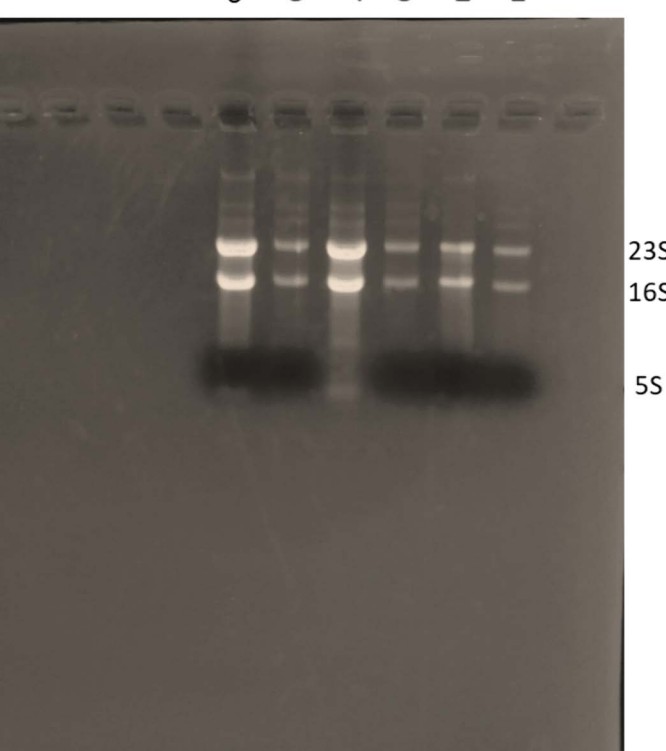

**Fig 1. Electrophoretic separation of _E. coli_ total RNA.**

## Influence of loading buffer components

Critically, and quite unexpectedly, clear and well-resolved separation of RNA bands was observed in the borax gel (Fig 1, Lanes 1–6) even when samples were loaded without the BPB tracking dye (Colorless Buffer, Lane 4) or without the denaturant SDS (Lanes 2 & 3), suggesting the borax medium itself possesses sufficient denaturing capability. The clear separation of _E. coli_ rRNA even when using simplified loading buffers strongly supports the hypothesis that borax facilitates RNA denaturation, a function traditionally reserved for harsh chemicals. This ability significantly simplifies the overall RNA analysis protocol. Furthermore, analysis revealed a significant advantage in visualization: the small ribosomal RNA (5S rRNA band) was noticeably enhanced and sharper when using the colorless Buffer (Lane 4) compared to formulations containing the bromophenol blue tracking dye. This finding indicates that BPB may interfere with the fluorescence and resolution of the fastest-migrating small RNA species. This leads to a key practical recommendation: for applications demanding optimal visualization of the smallest RNA fragments, a zero-BPB loading buffer (Colorless Buffer) is recommended. Conversely, for users less familiar with gel loading or when high-throughput loading is required, a minimal concentration of BPB may be used as a compromise to aid tracking while minimally impacting the 5S visualization.

Total _E. coli_ RNA was separated by Borax Agarose Gel Electrophoresis to investigate the influence of various loading buffer formulations on migration patterns. The 0.8% agarose gel was prepared with ethidium bromide (EtBr) integrated prior to casting. Electrophoresis was conducted at 120V for 25minutes. Lanes 1, 5, & 6: Total RNA loaded using a standard bromophenol blue loading buffer; lanes 2 & 3: Total RNA loaded using the standard buffer with Sodium Dodecyl Sulfate (SDS) omitted; lane 4: Total RNA loaded using a colorless buffer, with bromophenol blue omitted.

## Alternative approaches and concentration-dependent effects

While formaldehyde is the standard denaturant for RNA electrophoresis, alternative approaches have been explored for RNA integrity assessment. For instance, Aranda and co-workers described a "bleach gel" approach using sodium hypochlorite for analyzing RNA quality [14]. This method has had a significant impact on the field, likely due to sodium hypochlorite's ability to inactivate RNases, thereby preserving RNA integrity. Furthermore, hydrogen peroxide has also been explored as an alternative for RNA analysis in agarose gels [15], possibly acting as an oxidizing agent contributing to RNA stabilization. To investigate the concentration-dependent effects of borax on RNA integrity, RNA samples were treated with a range of borax concentrations (50 µM - 10 mM). It was observed that higher concentrations of borax (2–10 mM) caused degradation to RNA (S1 Fig), likely due to its chaotropic properties at these elevated levels. However, lower concentrations (50–300 µM) did not significantly affect RNA integrity when run on TAE gels (Fig 2), indicating that borax at these lower concentrations is not inherently destructive to RNA. The 23S:16S rRNA ratio appeared clearer in TAE gels where samples were pre-treated with different concentrations of borax (Fig 2), whereas the overall separation and sharpness of RNA bands were superior in borax gels (Figs 1 and 3). This suggests that borax can effectively separate RNA molecules at lower concentrations without compromising RNA integrity.

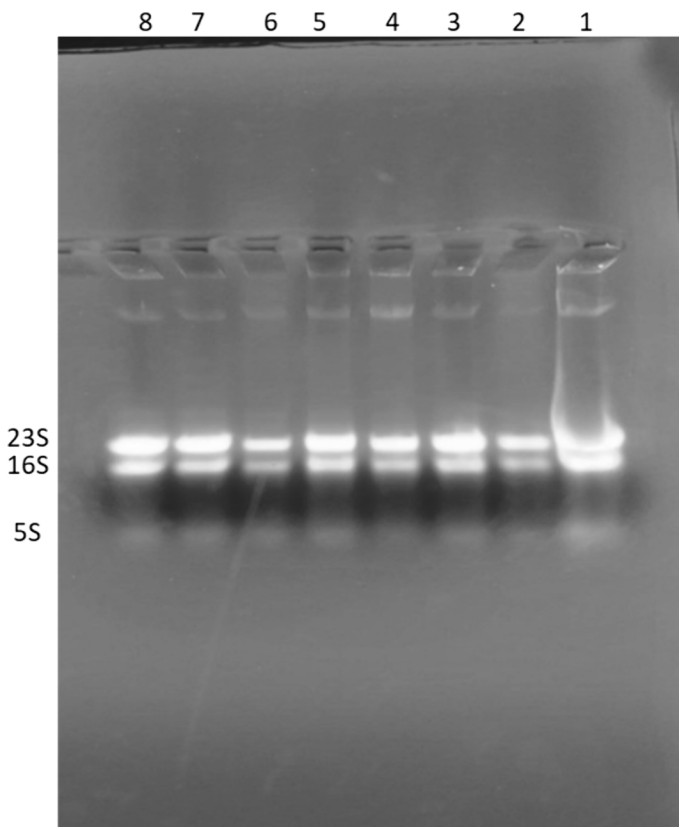

**Fig 2. Effect of low borax concentration pre-treatment on total *E. coli* RNA integrity in TAE agarose gel.** Total *E. coli* RNA samples were incubated with a range of low borax concentrations and subsequently separated using a standard TAE agarose gel electrophoresis system. Lane 1: RNA sample treated with 0 µM borax (Untreated Control). Lanes 2–7: RNA samples treated with borax concentrations of 50 µM, (Lane 2), 100 µM, (Lane 3) 150 µM, (Lane 4), 200 µM (Lane 5), 250 µM (Lane 6), and 300 µM (Lane 7).

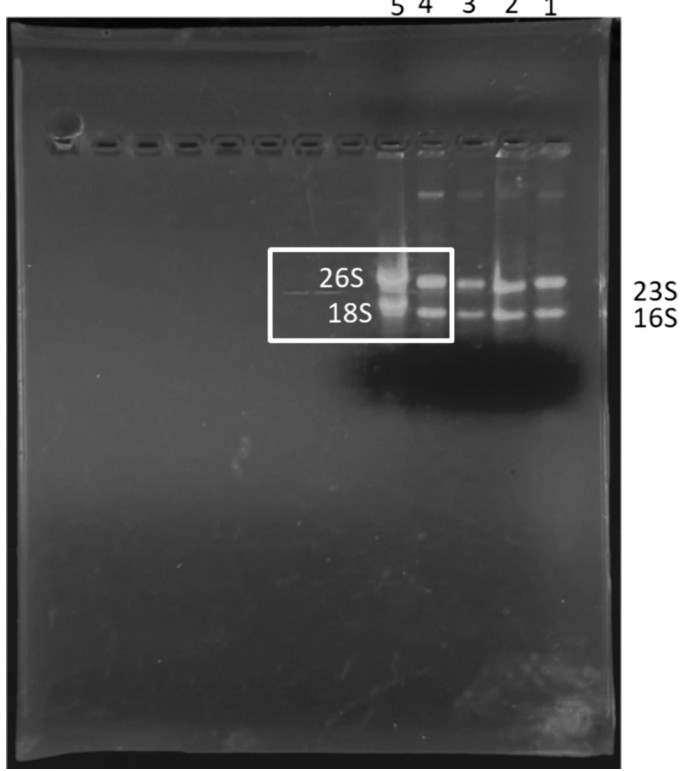

**Fig 3. Universal applicability of borax-based gel electrophoresis to diverse species the figure demonstrates the successful separation and assessment of total RNA integrity extracted from diverse microbial species using the 5 mM borax agarose gel system.** The method effectively resolves both prokaryotic ribosomal RNA (23S and 16S) and eukaryotic ribosomal RNA (26S/28S and 18S) species, confirming its utility across biological kingdoms. The RNA samples were loaded as follows.

## Universal applicability: Separation of eukaryotic and prokaryotic RNA

To address the fundamental question of universal applicability, the borax method was applied to total RNA from diverse species. This demonstrated the method's capacity to resolve both prokaryotic RNA (distinct 23S and 16S rRNA bands in bacteria) and eukaryotic RNA (clear 26S/28S and 18S rRNA bands in *Candida* species), as shown in Fig 3. The ability of the borax buffer to effectively separate these larger eukaryotic species (26S/28S) is highly significant, as it confirms the borax-based system provides the necessary environment to fully resolve complex RNA secondary structures, a characteristic previously limited to harsh denaturing conditions (e.g., formaldehyde). This finding validates the use of borax as a universal, safer alternative to formaldehyde for comprehensive RNA integrity assessment across different biological kingdoms.

- Lane 1: Total RNA from the Gram-negative bacterium *Pseudomonas aeruginosa*.

- Lane 2: Total RNA from the Gram-positive bacterium *Staphylococcus aureus*.

- Lane 3: Total RNA from the Gram-positive bacterium *Enterococcus faecalis*.

- Lane 4: Total RNA from the eukaryotic fungus *Candida glabrata*.

- Lane 5: Total RNA from the eukaryotic fungus *Candida albicans*.

## Mechanism of RNA separation in borax gels

While the precise mechanism of RNA separation in borax gels requires further investigation, our findings provide strong evidence that borax contributes to RNA separation by maintaining a stable pH and ionic environment within the gel matrix, thereby minimizing RNA aggregation and promoting uniform migration [21]. Furthermore, our observations strongly suggest that borax possesses a subtle yet effective denaturing-like property, capable of unfolding RNA secondary structures to allow for accurate size-based separation comparable to methods employing strong denaturants. This property is distinct from typical non-denaturing buffers, which may not fully resolve complex RNA structures crucial for integrity assessment. The successful subcloning of plasmid DNA extracted from a borax gel further demonstrates the compatibility of this method with downstream applications, suggesting that borax does not irreversibly damage nucleic acids.

## Future research

While this study provides a robust demonstration of the borax method across six microbial species, future research should focus on:

1. Further studies are needed to fully elucidate the mechanism of borax's denaturing-like activity.

2. Additionally, investigating the long-term stability of RNA in borax gels would be valuable.

3. Finally, a direct comparison of RNA integrity assessed by borax gels with established traditional methods (e.g., Agilent Bioanalyzer) and investigating the compatibility of borax gels with different RNA isolation methods would significantly strengthen the validation of this novel approach and potentially broaden its utility.

## Conclusion

This study decisively demonstrates the feasibility and unexpected efficacy of using borax as a buffer for RNA electrophoresis, offering a compelling and safer alternative to traditional formaldehyde-based methods. The borax-based method presents several distinct advantages, including enhanced safety, a simplified protocol, and reduced electrophoresis time. While further research is needed to fully understand the underlying mechanisms, these preliminary findings suggest that borax-based gels can be a valuable, accessible, and robust tool for RNA integrity assessment in various research and clinical settings. By eliminating the need for hazardous chemicals and cumbersome pre-treatment steps, the borax-based method significantly enhances safety and efficiency, making it a more accessible technique for routine RNA analysis. Future studies will be essential to further validate this approach and explore its broader applications in RNA research and diagnostics.

## Supporting information

**S1 Fig. Concentration dependent effects of borax on RNA integrity.**
(TIF)

**S1 File. Raw images.** Original uncropped and unadjusted blot/gel images.
(PDF)

## Acknowledgments

The author is grateful to Mr. Shamsi A. Shamsi for providing all bacterial strains (*Escherichia coli*, *Pseudomonas aeruginosa*, *Staphylococcus aureus*, and *Enterococcus faecalis*), and to Miss Huda I. Azaga for providing the eukaryotic fungal strains (*Candida glabrata* and *Candida albicans*).

## Author contributions

**Conceptualization:** Abdulhadi Albaser.

**Data curation:** Abdulhadi Albaser.

**Formal analysis:** Abdulhadi Albaser.

**Investigation:** Abdulhadi Albaser.

**Methodology:** Abdulhadi Albaser.

**Project administration:** Abdulhadi Albaser.

**Resources:** Abdulhadi Albaser.

**Supervision:** Abdulhadi Albaser.

**Validation:** Abdulhadi Albaser.

**Visualization:** Abdulhadi Albaser.

**Writing – original draft:** Abdulhadi Albaser.

**Writing – review & editing:** Abdulhadi Albaser.

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
