## [Decision Letter · Decision Letter 0]

5 Nov 2025

Dear Dr. Albaser,

Thank you for submitting your manuscript to PLOS ONE. After careful consideration, we feel that it has merit but does not fully meet PLOS ONE’s publication criteria as it currently stands. Therefore, we invite you to submit a revised version of the manuscript that addresses the points raised during the review process.

**1) Include other types of RNA, such as eukaryotic RNA, in vitro synthesized RNA or others in the analysis;**
**2) Please, see the comments raised by the both reviewers.**

We look forward to receiving your revised manuscript.

Kind regards,

Paulo Lee Ho, Ph.D.

Academic Editor

PLOS ONE

**Journal Requirements:**

1. When submitting your revision, we need you to address these additional requirements. Please ensure that your manuscript meets PLOS ONE's style requirements, including those for file naming. The PLOS ONE style templates can be found at https://journals.plos.org/plosone/s/file?id=wjVg/PLOSOne_formatting_sample_main_body.pdf and https://journals.plos.org/plosone/s/file?id=ba62/PLOSOne_formatting_sample_title_authors_affiliations.pdf 2. We note that you have included the phrase “data not shown” in your manuscript. Unfortunately, this does not meet our data sharing requirements. PLOS does not permit references to inaccessible data. We require that authors provide all relevant data within the paper, Supporting Information files, or in an acceptable, public repository. Please add a citation to support this phrase or upload the data that corresponds with these findings to a stable repository (such as Figshare or Dryad) and provide and URLs, DOIs, or accession numbers that may be used to access these data. Or, if the data are not a core part of the research being presented in your study, we ask that you remove the phrase that refers to these data. 3. PLOS ONE now requires that authors provide the original uncropped and unadjusted images underlying all blot or gel results reported in a submission’s figures or Supporting Information files. This policy and the journal’s other requirements for blot/gel reporting and figure preparation are described in detail at https://journals.plos.org/plosone/s/figures#loc-blot-and-gel-reporting-requirements and https://journals.plos.org/plosone/s/figures#loc-preparing-figures-from-image-files. When you submit your revised manuscript, please ensure that your figures adhere fully to these guidelines and provide the original underlying images for all blot or gel data reported in your submission. See the following link for instructions on providing the original image data: https://journals.plos.org/plosone/s/figures#loc-original-images-for-blots-and-gels.   In your cover letter, please note whether your blot/gel image data are in Supporting Information or posted at a public data repository, provide the repository URL if relevant, and provide specific details as to which raw blot/gel images, if any, are not available. Email us at plosone@plos.org if you have any questions. 4. If the reviewer comments include a recommendation to cite specific previously published works, please review and evaluate these publications to determine whether they are relevant and should be cited. There is no requirement to cite these works unless the editor has indicated otherwise. 

Reviewers' comments:

**Comments to the Author**

1. Is the manuscript technically sound, and do the data support the conclusions?

Reviewer #1: Partly

Reviewer #2: Yes

2. Has the statistical analysis been performed appropriately and rigorously?

Reviewer #1: N/A

Reviewer #2: N/A

3. Have the authors made all data underlying the findings in their manuscript fully available?

Reviewer #1: Yes

Reviewer #2: Yes

4. Is the manuscript presented in an intelligible fashion and written in standard English?

Reviewer #1: Yes

Reviewer #2: Yes

**Reviewer #1:**  The article presents a non-toxic alternative for RNA integrity analysis using borax. This type of approach is always a good path for the development of techniques that reduce complications associated with the use of classical methods in molecular biology. However, the article shows little innovation, being more of a technical report than a paper focused on innovative science.

I believe the findings could be substantially expanded through the use of different types of RNA. In the current study, the only RNA used was extracted from E. coli. Including other types of RNA, such as eukaryotic RNA, in vitro synthesized RNA, and circular RNAs, could broaden the scope and purpose of the article.

In this context, I recommend making improvements to the study and submitting it to journals with a more technical profile.

**Reviewer #2:** The paper by Abdulhadi Albaser makes very significant advances by (1) defining an

novel method for RNA analysis, (2) providing evidence for their use and application in RNA study, and (3) show an enhanced safety, simplified protocols, and reduced electrophoresis time method

The manuscript is well written and the work overall sound, but a few issues should be addressed as described below:

1- Add the method to obtain total RNA samples from E. coli

2- In “Electrophoresis - Approximately 20 μL of RNA", what would be the RNA concentration in μg used?

3- In figure 1, add lane1 and 2

4- In Figure 3: "Lanes 2-7: RNA samples treated with different concentrations of borax." What is the concentration of borax tested in each lane?

**Do you want your identity to be public for this peer review?** For information about this choice, including consent withdrawal, please see our Privacy Policy

Reviewer #1: **Yes:** Wesley Luzetti Fotoran

Reviewer #2: **Yes:** Felipe Chambrgo

---

## [Author Response · Author response to Decision Letter 1]

10 Dec 2025

Rebuttal Letter (Response to Reviewers)

Title: Borax-Based Gel Electrophoresis: A Novel Approach for RNA Integrity Analysis Manuscript Number: PONE-D-25-34099 (Placeholder)

Dear Dr. Ho and Reviewers,

I thank you and the reviewers for the insightful and constructive comments on my manuscript. I appreciate the time taken for the careful review of my work and am pleased to submit a revised version that addresses all the concerns raised by the Academic Editor and the reviewers.

I believe the inclusion of new data on eukaryotic RNA and the detailed clarification of the loading buffer study significantly strengthen the paper, particularly its claim of universal applicability and simplified methodology.

Below is a detailed, point-by-point response to each comment. All changes I made have been highlighted in the manuscript titled 'Revised Manuscript with Track Changes.docx'.

Academic Editor & Journal Requirements

Req. # Original Comment My Response

AE #1 Include other types of RNA, such as eukaryotic RNA, in vitro synthesized RNA or others in the analysis. I agree this expansion was crucial for demonstrating the method's versatility. I performed new experiments and included data demonstrating the successful application of the borax method to total RNA extracted from two eukaryotic fungal species (Candida glabrata and Candida albicans), alongside the prokaryotic species. This new data confirms the borax buffer's capacity to resolve complex eukaryotic secondary structures (26S/28S and 18S bands). This information is presented in Figure 3 and discussed in the Results section under "Universal Applicability."

JR #2 We note that you have included the phrase “data not shown” in your manuscript. I have fully addressed this. I replaced the single instance of "data not shown" with a citation to a new Supporting Information file, which contains the relevant data: (S1 fig). This file will be submitted separately.

JR #3 PLOS ONE now requires that authors provide the original uncropped and unadjusted images underlying all blot or gel results. I confirm that all original, uncropped, and unadjusted raw images underlying the data in Figures 1, 2, 3, and S1 Fig have been compiled into a single file, which I will upload as Supporting Information titled 'Original Uncropped Images' to meet this requirement.

Reviewer #1

Comment # Original Comment My Response

1. The article shows little innovation, being more of a technical report than a paper focused on innovative science... I believe the findings could be substantially expanded through the use of different types of RNA (eukaryotic RNA, in vitro synthesized RNA, and circular RNAs). I appreciate this feedback and have significantly expanded the scope and innovative value of the work. As noted above (AE #1), the study now includes data demonstrating the successful resolution of eukaryotic RNA (26S/28S and 18S bands) from two different Candida species (Figure 3). I argue that demonstrating the capacity of a non-toxic, non-formaldehyde method to resolve complex eukaryotic RNA secondary structures constitutes a significant innovation. Additionally, the revised manuscript includes detailed methods and results for my loading buffer component experiment (Figure 1), which yielded a crucial practical discovery: that the absence of the BPB tracking dye significantly enhances the visualization of the 5S rRNA band, offering a clear optimization recommendation for users of this novel technique.

Reviewer #1

Comment # Original Comment My Response

1. Add the method to obtain total RNA samples from E. coli. Done. I have clarified the sample preparation protocol in the "Sample Preparation and RNA Extraction" section of the Methods to confirm that the identical acidic lysis protocol was applied to all species, including E. coli.

2. In “Electrophoresis - Approximately 20 µL of RNA", what would be the RNA concentration in mg used? Done. I have added the required concentration detail to the "Electrophoresis" section of the Methods, which now reads: "Approximately 20 µL (containing approximately 1.5-2.0 µg of total RNA) was mixed with..."

3. In figure 1, add lane 1 and 2. Done. I have fully revised the Figure 1 Legend to provide an explicit, lane-by-lane breakdown of all samples shown (Lanes 1-6), clearly differentiating between the Standard, SDS-Omitted, and Colorless (BPB -Omitted) loading buffer formulations used.

4. In Figure 3: "Lanes 2-7: RNA samples treated with different concentrations of borax." What is the concentration of borax tested in each lane? Done. This figure has been renumbered to Figure 2 in the revised manuscript. I have added a detailed legend for Figure 2 that explicitly lists the borax concentration tested in each lane (Lanes 1-7: 0 mM up to300 µM) to ensure full reproducibility.

I trust that the revised manuscript and these detailed responses have adequately addressed all concerns. I am grateful for the opportunity to revise and resubmit my work to PLOS ONE.

Sincerely,

Abdulhadi Albaser

---

## [Decision Letter · Decision Letter 1]

17 Feb 2026

Borax-Based Gel Electrophoresis: A Novel Approach for RNA Integrity Analysis

PONE-D-25-34099R1

Dear Dr. Albaser,

We’re pleased to inform you that your manuscript has been judged scientifically suitable for publication and will be formally accepted for publication once it meets all outstanding technical requirements.

Kind regards,

Paulo Lee Ho, Ph.D.

Academic Editor

PLOS One

Additional Editor Comments (optional):

Reviewers' comments:

Reviewer's Responses to Questions

**Comments to the Author**

Reviewer #2: All comments have been addressed

2. Is the manuscript technically sound, and do the data support the conclusions?

Reviewer #2: Yes

3. Has the statistical analysis been performed appropriately and rigorously?

Reviewer #2: N/A

4. Have the authors made all data underlying the findings in their manuscript fully available?

Reviewer #2: Yes

5. Is the manuscript presented in an intelligible fashion and written in standard English?

Reviewer #2: Yes

Reviewer #2: The revised version answers all my questions. The figures are appropriate and the captions are well described.

**Do you want your identity to be public for this peer review?** For information about this choice, including consent withdrawal, please see our Privacy Policy

Reviewer #2: **Yes:** Felipe S Chambergo

---

## [Editor Report · Acceptance letter]

PONE-D-25-34099R1

PLOS One

Dear Dr. Albaser,

I'm pleased to inform you that your manuscript has been deemed suitable for publication in PLOS One. Congratulations! Your manuscript is now being handed over to our production team.

Kind regards,

on behalf of

Dr. Paulo Lee Ho

Academic Editor

PLOS One